# Investigating the dose-dependency of the midgut escape barrier using a mechanistic model of within-mosquito dengue virus population dynamics

**Rebecca M. Johnson[1]ʘ, Isaac J. Stopard[2]ʘ, Helen M. Byrne[3], Philip M. Armstrong[1]‡, Douglas E. Brackney[1]‡, Ben Lambert[4]‡ ***

**1** Center for Vector-Borne and Zoonotic Diseases, Department of Entomology, The Connecticut Agricultural Experiment Station, New Haven, Connecticut, United States of America, **2** MRC Centre for Global Infectious Disease Analysis, School of Public Health, Faculty of Medicine, Imperial College London, London, United Kingdom, **3** Mathematical Institute, University of Oxford, Oxford, United Kingdom, **4** Department of Statistics, University of Oxford, Oxford, United Kingdom

ʘ These authors contributed equally to this work.
‡These authors also contributed equally to this work.
* ben.c.lambert@gmail.com

**Data Availability Statement:** All software and data to reproduce this work are available in a Zenodo repository (DOI: 10.5281/zenodo.10835569).

## Abstract

Arboviruses can emerge rapidly and cause explosive epidemics of severe disease. Some of the most epidemiologically important arboviruses, including dengue virus (DENV), Zika virus (ZIKV), Chikungunya (CHIKV) and yellow fever virus (YFV), are transmitted by *Aedes* mosquitoes, most notably *Aedes aegypti* and *Aedes albopictus*. After a mosquito blood feeds on an infected host, virus enters the midgut and infects the midgut epithelium. The virus must then overcome a series of barriers before reaching the mosquito saliva and being transmitted to a new host. The virus must escape from the midgut (known as the midgut escape barrier; MEB), which is thought to be mediated by transient changes in the permeability of the midgut-surrounding basal lamina layer (BL) following blood feeding. Here, we present a mathematical model of the within-mosquito population dynamics of DENV (as a model system for mosquito-borne viruses more generally) that includes the interaction of the midgut and BL which can account for the MEB. Our results indicate a dose-dependency of midgut establishment of infection as well as rate of escape from the midgut: collectively, these suggest that the extrinsic incubation period (EIP)—the time taken for DENV virus to be transmissible after infection—is shortened when mosquitoes imbibe more virus. Additionally, our experimental data indicate that multiple blood feeding events, which more closely mimic mosquito-feeding behavior in the wild, can hasten the course of infections, and our model predicts that this effect is sensitive to the amount of virus imbibed. Our model indicates that mutations to the virus which impact its replication rate in the midgut could lead to even shorter EIPs when double-feeding occurs. Mechanistic models of within-vector viral infection dynamics provide a quantitative understanding of infection dynamics and could be used to evaluate novel interventions that target the mosquito stages of the infection.

**Funding:** This work was supported in part by grants to DG and PA from the National Institutes of Health, National Institute of Allergy and Infectious Diseases (AI148477). The funders had no role in study design, data collection and analysis, decision to publish, or preparation of the manuscript.

## Author summary

*Aedes* mosquitoes are the main vectors of dengue virus (DENV), Zika virus (ZIKV) and yellow fever virus (YFV), all of which can cause severe disease in humans with dengue alone infecting an estimated 100–400 million people each year. Understanding the processes that affect whether, and at which rate, mosquitoes may transmit such viruses is, hence, paramount. Here, we present a mathematical model of virus dynamics within infected mosquitoes. By combining the model with novel experimental data, we show that the course of infection is sensitive to the initial dose of virus ingested by the mosquito. The data also indicate that mosquitoes which blood feed subsequent to becoming infected may be able to transmit infection earlier, which is reproduced in the model. This is important as many mosquito species feed multiple times during their lifespan, and any reduction in time to dissemination will increase the number of days that a mosquito is infectious and so enhance the risk of transmission. Our study highlights the key and complementary roles played by mathematical models and experimental data for understanding within-mosquito virus dynamics.

## Introduction

Dengue virus (DENV), Zika virus (ZIKV), chikungunya (CHIKV), West Nile virus (WNV), and yellow fever virus (YFV) represent some of the most prevalent and deadly arthropod-borne (arbo)viruses, with DENV alone accounting for an estimated 100–400 million cases annually [1]. DENV and many other flaviviruses are primarily transmitted by *Aedes aegypti* (*Ae. aegypti*) and, to a lesser extent, by *Ae. albopictus*. While DENV is endemic in more than 100 countries and causes sporadic but devastating epidemics, other flaviviruses such as ZIKV have emerged more suddenly [1]. It is estimated that ZIKV was introduced into Brazil in 2013–2014, with the first cases being detected in 2015 [2, 3]. A subsequent rapid range expansion and epidemic led to millions of infections, infants born with microcephaly, and the spread of ZIKV to 89 countries as of 2023 [3–5]. These arboviral epidemics have huge impacts on public health and may become more prevalent as climate change and urbanization result in the expansion of habitats that are suitable for key vectors [6]. Thus, understanding the dynamics that drive transmission and maintenance of such flaviviruses is important for predictive modelling and future control measures.

The *Aedes* mosquitoes that transmit many of these arboviruses thrive in urban environments and are aggressive, daytime-biters that have been shown to take 0.63–0.76 blood meals on average per day in Thailand and Puerto Rico, with many taking multiple blood meals per gonotrophic cycle [7, 8]. It has previously been shown that giving mosquitoes a second non-infectious blood meal several days following an initial infectious feed leads to earlier dissemination and, subsequently, a shorter extrinsic incubation period (EIP) in *Ae. aegypti* and *Ae. albopictus* mosquitoes that are infected with DENV, ZIKV and CHIKV [9]. The same phenomenon has been observed for *Anopheles gambiae* infected with *Plasmodium falciparum*, although the mechanism appears to be related to nutritional advantages that reduce competition among parasites and lead to faster parasite growth rates [10, 11]. Whereas, in mosquitoes infected with viruses, mechanical damage to the basal lamina caused by midgut expansion seems to play a larger role [9, 12]. Mosquitoes in the field are thought to have short lifespans (10 days or less [13]), and only a small number are infected at any given time. This means that any reduction in the extrinsic incubation period (EIP), the time between virus ingestion and

its presence in the saliva, can dramatically impact the number of days that a mosquito is infectious and the chance that it transmits virus to a new host [14]. Therefore, understanding the processes which influence the rate at which mosquitoes become infectious is critical to quantifying transmission risk.

The dose of virus ingested can also impact the rate of midgut infection and the EIP with higher blood meal titers of DENV serotype 2 (DENV-2), consistently leading to higher midgut infection rates in diverse *Ae. aegypti* populations collected across Mexico and the United States [15]. Other studies have shown that higher DENV-2 or DENV serotype 3 (DENV-3) infection titers reduce the EIP even in the presence of pathogen-blocking wMel *Wolbachia* [16, 17]. Similarly, when DENV-infected patients were used to feed mosquitoes, mosquito infection rate increased with patient DENV plasma titer. Furthermore, higher DENV plasma titers lead to a higher viral load in mosquitoes and a larger proportion of mosquitoes with disseminated infection and virus in their saliva at 14 days post infection [18]. While in this study we focus on studying the impact of DENV-2 concentration on dissemination in *Ae. aegypti*, similar interactions between viral titer and midgut infection and saliva transmission have been seen with ZIKV in *Ae. aegypti* and *Ae. albopictus* [19–21]. Most studies measure dissemination as the proportion of all blood-fed mosquitoes with disseminated infection making it impossible to separate the impact of changes to the bloodmeal titer on infection and subsequent dissemination. Some studies do, however, measure dissemination as the proportion of mosquitoes with infected midguts which have disseminated infection [21, 22], and here we follow suit. These studies have been used to build mathematical models for the impact of ZIKV titer on $R_0$ and the epidemic potential of less competent ZIKV vectors such as *Ae. albopictus* [21, 22].

Mathematical modelling is a complementary tool to laboratory experiments: i) it can be used to perform virtual experiments, which are difficult and/or expensive to conduct on real mosquitoes; ii) it can provide quantitative predictions which can be tested by experiments; and, iii) it can provide a mechanistic basis for interpreting experimental data which is often clouded by considerable measurement uncertainty. Mathematical models have been an important tool in understanding mosquito-borne pathogen epidemiology and are now routinely used to inform policy [23–26]. They have also been extensively used to explore within-host pathogen dynamics [22, 27–29]. Despite this work exploring within-host pathogen dynamics, only a limited number of mathematical models have explored viral dynamics in the mosquito [22, 29–31], and the majority of existing within-vector modelling work has so far focused on malarial parasites [32–36]. This work has, however, hinted that these models can uncover biological insights which are difficult to intuit from experimental data alone. Childs and Prosper (2020), for example, developed an ordinary differential equation (ODE) model of *Plasmodium berghei* dynamics within *Anopheles* mosquitoes, which explored the hypothesis that a small proportion of oocysts (a life stage of malarial parasites) burst to yield sporozoites [34]. Another study modelled *Plasmodium falciparum* dynamics within *Anopheles* mosquitoes and demonstrated how accounting for mosquito mortality and parasite development processes were important for estimating the EIP [35].

In this paper, we introduce a mechanistic model of within-mosquito viral dynamics and fit the model to experimental data collected by infecting *Ae. aegypti* mosquitoes with DENV. Despite being relatively simple, our model provides quantitative predictions which are consistent with our experimental data about how increasing the infection dose influences the rate at which mosquitoes become infectious—a phenomenon that has been demonstrated in previous experimental studies [21, 22] and modelling work [29]. It also provides a framework that could be used to explore how perturbations to key mechanisms, for example, due to viral mutations or environmental fluctuations, may affect transmission rates. Importantly, our model shows that allowing the rate at which the virus can escape from the midgut to depend

on the physical integrity of the encompassing basal lamina layer can explain why mosquitoes bloodfed on a second noninfectious bloodmeal may hasten dissemination rates—a hypothesis previously suggested by Armstrong et al. [9]. Simulations from our model indicate that there is a sweet spot in parameter space, where this double-feeding effect is strongest, so small changes in the parameter values may lead to substantial changes in the importance of repeated feeding on transmission. Our model is a step towards developing a higher fidelity model of viral infection dynamics within mosquitoes and, more generally, demonstrates the utility of collaborations between vector biologists and mathematical modellers in advancing understanding of within-vector viral dynamics.

## Methods and data

### Mosquitoes, viruses, cell culture

All experiments were conducted using *Ae. aegypti* (Orlando strain, collected from Orlando, Florida in 1952) mosquitoes that were maintained at 27˚C with a 14:10 light:dark cycle and fed on defibrinated sheep's blood using a glass water-jacketed membrane feeder. Larvae were reared at a concentration of approximately 200 larvae per liter of water and fed with a (3:2) mix of liver powder:yeast. Pupae were strained and adults emerged within screened cages. Adult mosquitoes were fed 10% sugar *ad libitum* and starved overnight before blood feeding.

*Ae. albopictus* C6/36 cells were used to grow virus for these studies. Cells were maintained through passaging at a 1:10 dilution with minimum essential medium containing 10% FBS, 1X non-essential amino acids (Gibco, Grand Island, NY, 11140050), 1X antibiotic-antimycotic (Gibco, Grand Island, NY, 15240062), and sodium bicarbonate. To amplify virus, cells were grown to 70–80% confluency in 75-cm2 flasks, 250 μl DENV-2 virus (125270/VENE93; Gen-Bank:U91870) stocks were diluted in 3 ml of media and overlaid on cells on a rocking platform at room temperature for 1 h. Following incubation, total media volume was brought up to 15 ml and infected cells were incubated at 28˚C with 5% $CO_2$ for 5 days. Virus-containing supernatant was diluted in defibrinated sheep's blood before feeding to mosquitoes and an aliquot was frozen in -80˚C for each experiment and dilution.

### Infection and dissemination assays

Adult female mosquitoes that were approximately one week old were starved overnight and fed on varying concentrations of DENV-2. For initial experiments, 1:1, 1:5, or 1:12 dilutions of DENV-2 in defibrinated sheep's blood were offered to mosquitoes and engorged mosquitoes were sorted into cups. Mosquito midguts and legs were harvested at 3, 5, and 8 days post-infection (DPI) to assess infection and dissemination rates and to determine viral titers (see Fig 1A). The dissections provided titer data across the time course of the experiment; these data were either continuous and represented viral titers if the measured levels were above the limits of detection, or were zeros representing measurements below this threshold. Due to this thresholding behaviour, we collectively refer to these data as being "semi-continuous". Additional experiments quantified infection rates using dichotomous measures (binary outcome: infected or not-infected) 5 DPI for mosquitoes fed 1:1, 1:5, 1:12, 1:20, or 1:25 dilutions of DENV-2 in defibrinated sheep's blood (Fig 1B). For time course experiments measuring the effect of single and double feeding on dissemination, mosquitoes were fed a 1:5 dilution of DENV-2 and the double-fed group was given a second blood meal three days later. Mosquito bodies and legs were harvested at 4, 5, 6, 7, 8, and 12 DPI for both single and double-fed groups and assessed in a dichotomous manner (Fig 1C).

Mosquito midguts, bodies, and legs were extracted using a Mag-Bind Viral DNA/RNA 96 Kit (Omega Bio-tek Inc., Norcross, GA) and a Kingfisher Flex automated nucleic acid

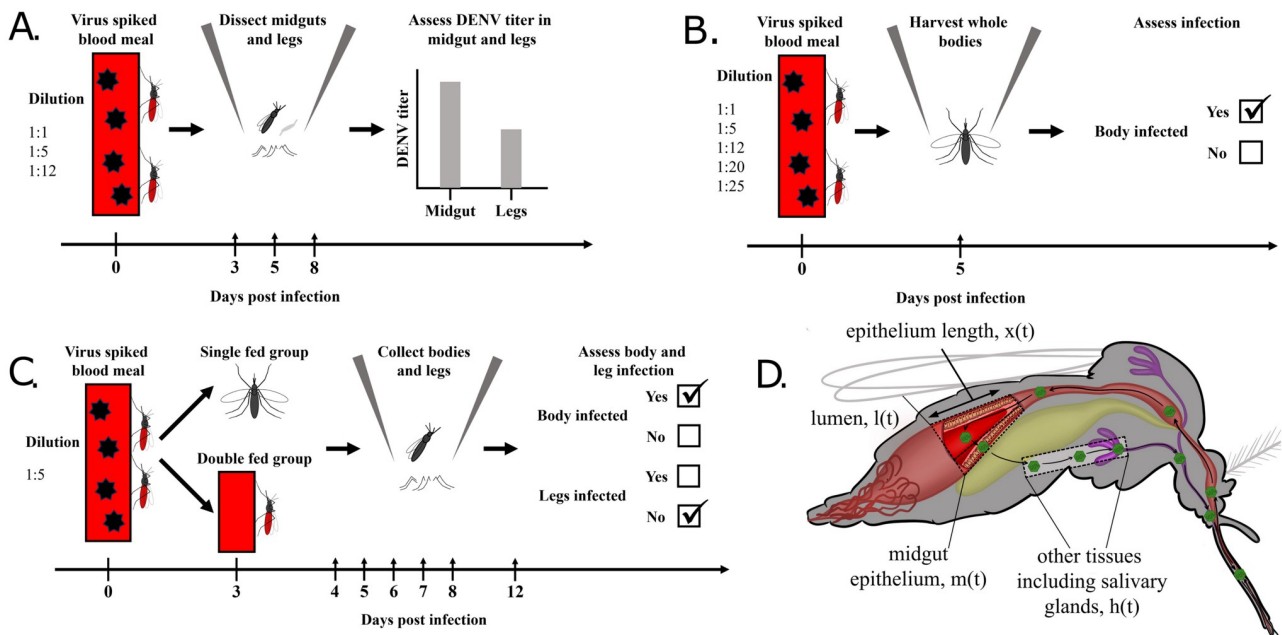

**Fig 1. Schematic of experiments (A, B and C) and of the mosquito including the compartments of our mathematical model (D).** Once ingested as part of the bloodmeal, the virus is in the mosquito lumen. It subsequently invades the midgut epithelium before escaping into the haemocoel, invading secondary tissues such as the salivary glands, and being transmitted during a subsequent feeding episode. The rate of invasion of the midgut and the rate of viral escape from it are controlled by the epithelium length.

extraction device (ThermoFisher Scientific, Waltham, MA) as per manufacturer instructions. Samples were assessed for DENV-2 titers or infection presence / absence using qPCR and an iTaq Universal Probes One-Step Kit (BioRad, 1725141). Reaction volumes were 20 μl and previously developed DENV-2 Fwd: CATGGCCCTKGTGGCG and DENV-2 Rev: CCCCATC-TYTTCAGTATCCCTG primers were used with DENV-2 Probe: [FAM] TCCTTCGTTTCCTAACAATCC [BHQ1] [37]. Conditions were 50°C for 30 min and 95°C for 10 min followed by 40 rounds of 50°C for 15 sec and 60°C for 1 min. For continuous data, a standard curve was used to quantify viral titer and genome equivalents/tissue (GE/tissue) were calculated. For dichotomous data, no standard curve was used and the presence of infection was scored using a qPCR Ct cutoff value of 36 with positive and negative control wells to determine assay effectiveness.

## Viral dilution quantification

Initial viral concentrations used to feed mosquitoes were measured via a focus forming assay. C6/36 cells were passaged and 96-well plates were seeded with $3 \times 10^5$ cells per well. The following day, viral dilutions used to feed mosquitoes were thawed and diluted via serial dilution in serum free media. Media were removed from 96-well plates and 30 μl of each serial dilution was placed onto wells. Plates were incubated at 28°C with 5% $CO_2$ for 1 hr before infection media were removed and wells were covered with 100 μl of 0.8% methylcellulose in minimum essential medium containing 10% FBS, 1X non-essential amino acids, 1X antibiotic-antimycotic, and sodium bicarbonate. Plates were incubated for 3 days at 28°C with 5% $CO_2$. Cells were fixed with 100 μl of 4% formaldehyde per well for 15 minutes at room temperature, washed 3x with PBS, and then permeabilized with 50 μl of 0.2% triton-X in PBS for 10 minutes at room

temperature. Cells were washed 3X with PBS and incubated overnight at 4˚C with 30 µl of flavivirus group antigen antibody D1–4G2-4–15 (4G2)(Novus Biologicals, NBP2–52709) diluted 1:100 in PBS. The following day, primary antibody was removed and plates were washed 3x with PBS. Donkey anti-mouse IgG (H+L) highly cross-adsorbed secondary antibody, alexa fluor 555(ThermoFisher Scientific, A-31570) was diluted 1:200 in PBS and 30 µl/well was applied to the plate. Plates were incubated overnight at 4˚C, washed 3x with PBS and dried. Fluorescent foci were counted using a Zeiss Axio Vert A1 microscope and focus forming units/ml (FFU/ml) were calculated for each viral dilution fed to mosquitoes.

## Basal lamina collagen IV damage

Midgut basal lamina collagen IV damage was assessed following blood feeding using a collagen hybridizing peptide as described previously [9]. Briefly, mosquito midguts were dissected at various times post blood meal (15 minutes post blood meal and 24, 36, 48, 72, or 96 hours post blood meal) and pooled into groups of five. Unfed mosquito guts were dissected and pooled and served as a control for collagen IV damage. Midguts were fixed overnight in 2.5% glutaraldehyde and 2% paraformaldehyde, washed 3X with PBS, blocked with 5% bovine serum albumen in PBS, and incubated overnight at 4˚C with activated 5 µM 5-FAM conjugated collagen hybridization peptide (CHP). Midguts were washed 5X to remove excess peptide and bound CHP was removed by incubating midguts in 1 mg/ml elastase in PBS for 2 hr at 27˚C. The CHP-containing supernatant was removed, diluted 1:1 with PBS in a 96-well plate, and the level of CHP fluorescence was measured via a BioTek SYNERGY H1 microplate reader using the area scan feature and an excitation/emission of 485/515 nm.

## Mathematical model

Our chosen framework is a non-spatial compartmental model where the dynamics of virus quantities are described by a system of autonomous ordinary differential equations. The model tracks the viral titers in three different locations within the mosquito: in the lumen, the midgut and other tissues (Fig 1D). Here, *other* represents those tissues, including the legs, to which the virus disseminates after escaping the midgut.

After an infectious blood meal, the virus enters the lumen. The level of virus in the lumen, $l(t)$ at time $t$ post-infection, is assumed to be governed by the following ODE:

$$\frac{dl(t)}{dt} = -\gamma l(t) - \frac{l(t)}{a + l(t)} k_{lm} x(t), \tag{1}$$

where $\gamma > 0$ represents the rate of degradation of the virus by digestive enzymes; the second term describes the rate at which the virus invades the midgut epithelium. Here, $k_{lm} > 0$ determines the maximum rate at which invasion can occur, and $a > 0$ is the level of virus at which the rate of invasion is at half its maximum (when $l(t) = a$, the rate of invasion is $\frac{k_{lm}}{2} x$). The initial conditions are given by an unknown parameter: $l(0) = l_0 > 0$, which represents the level of virus ingested when blood feeding. In Eq (1), $x(t)$ represents the length of the midgut epithelium layer, which stretches in response to a blood feed. For mathematical simplicity, we take a phenomenological approach and model this layer as a one-dimensional tissue. Stretching of this layer exposes a higher surface area of the epithelium to the lumen, since the shape of the lumen changes dramatically subsequent to bloodfeeding [38], and we allow a faster rate of viral invasion to result through an increased rate of contact between virus and midgut epithelial cells. Since the mechanism by which viruses invade midgut cells remains, however, poorly understood, it would be useful to probe this hypothesis through future experimental work. Subsequent to a blood feed, the epithelium layer returns to its natural length, $x^*$, representing

its extent when no blood meal is present. The rate at which it contracts is modelled as follows:

$$\frac{dx(t)}{dt} = -\eta(x(t) - x^*),$$
(2)

which is closed by assuming $x(0) = x_0 > x^*$ is the length of the midgut layer immediately post-blood meal. For mosquitoes undergoing a second blood meal at time $t_r > 0$, we assume that the epithelium length is discontinuously reset to $x(t_r) = x_0$, that is, we assume that subsequent blood meals may be smaller than the first as we implicitly assume the mosquito feeds only until it is full. In reality, mosquitoes likely imbibe very heterogeneous blood meals, but, without a means to measure within-mosquito variation in blood meal size, this assumption represents a parsimonious null hypothesis.

After invading the midgut, the virus replicates, and the quantity of virus in the midgut, $m(t)$, evolves according to the following ODE:

$$\frac{dm(t)}{dt} = \frac{l(t)}{a + l(t)} k_{lm}x(t) + \alpha_m m(t)(1 - m(t)/\kappa_m) - k_{mh}(x(t) - x^*)m(t),$$
(3)

where the second term represents logistic growth of the viral population. Considering the influence of this term on the midgut viral population dynamics in isolation, the proliferation rate of virus within this tissue immediately subsequent to infection is given by $\alpha_m m(t)$—that is, the viral population grows exponentially. At later times, the rate of growth slows as viral levels reach a carrying capacity, $\kappa_m > 0$. We assume that $m(0) = 0$. It has been hypothesised that the distortion of the epithelium layer leads to basal lamina damage through which the virus can pass [9, 38]. Over time, as the blood meal is digested, the epithelium returns to its original length (as dictated by Eq (2)), and the basal lamina layer is repaired. The last term on the right-hand side of Eq (3) represents the rate of escape of virus from the midgut layer during this transient period, with $k_{mh}(x_0-x^*)$ representing the maximum rate at which individual virus particles can escape.

After the virus passes through the basal lamina layer, it enters other tissues, where it again replicates. The dynamics of the level of virus in these tissues, $h(t)$, are assumed to be governed by:

$$\frac{dh(t)}{dt} = k_{mh}(x(t) - x^*)m(t) + \alpha_h h(t)(1 - h(t)/\kappa_h),$$
(4)

where the second term on the right-hand side of Eq (4) represents logistic growth of the virus population in this tissue (with growth parameter $\alpha_h > 0$ and carrying capacity $\kappa_h > 0$). We assume that $h(0) = 0$. We term the level of virus in these other tissues as the level of *disseminated infection* which is measured by dissecting the mosquito legs.

## Statistical model

To link the mathematical model described in the previous section with experimental observations, a statistical model governing measurement processes is required. We neglect inter-specimen variation: that is, we assume that the viral dynamics and recovery of the epithelium across all mosquitoes are governed by Eqs (1)–(4), with the same set of parameter values. This assumption of homogeneity is necessary since observations were only made at a single time point for each specimen (when the mosquitoes were dissected), meaning we cannot recover parameter estimates for individual specimens. The dynamics are likely to be heterogeneous across specimens [39], but without time series for individual specimens, it is not straightforward to separate variation in underlying dynamics from inherent variation due to imperfect

measurement of viral titers. Recently, experimental techniques have emerged which allow such data on viral infection and titers to be collected in a continuous manner using mosquito excreta [40, 41]. These data correlate closely with viral dissemination and, such techniques, in combination with continuous saliva sampling as through honey-soaked nucleic acid preservation cards, could be used in future studies to quantify inter-specimen heterogeneity in viral dynamics [42].

Our statistical model must account for the two types of experimental data: semi-continuous measurements of the DENV viral titer in the midgut and in the legs / body (i.e. disseminated infection represented by the solution to Eq (4)), which are either zero if the DENV titer is below threshold for detection ($C_t > 36$) or the positive DENV titer value, if above the threshold ($C_t < 36$); and dichotomous measurements of infection presence / absence of the midgut and legs / body.

In what follows, we distinguish between modelled quantities (e.g. $m(t)$) and corresponding observations of these quantities from laboratory experiments (e.g. $\bar{m}(t)$). Considering first the semi-continuous midgut data, we assume that the observed infection data for the midgut, $\bar{m}(t)$, represents the outcome of two distinct processes: i) invasion of the midgut and ii) proliferation within it. Our data indicate presence / absence of midgut infection depends on the initial dose of virus ingested during blood meal, $l_0^{\{1\}}/d$, where $l_0^{\{1\}}$ represents the quantity of virus taken in for the 1:1 dilution, and $d \geq 1$ represents the dilution of that particular experiment. We allow the probability of successful invasion, $\phi(d)$, to depend on the dilution through a generalised logistic function:

$$\phi(d) = \frac{1}{1 + q \exp(b \log d)}, \tag{5}$$

where $q > 0$ and $b > 0$ determine the dose-response behaviour for more moderate concentrations.

For any observations, $\bar{m}(t)$, we assume the probability density is given by:

$$p(\bar{m}(t)|d) = \begin{cases} \phi(d)\text{log-normal}_\theta(\bar{m}(t)|\log m(t|d), \sigma_m), & \text{if } \bar{m}(t) > \theta, \\ (1 - \phi(d)) + \phi(d)\text{F}(\theta|\log m(t|d), \sigma_m), & \text{if } \bar{m}(t) = 0, \\ 0, & \text{otherwise,} \end{cases} \tag{6}$$

where $\theta > 0$ represents the detection limit of our experimental instruments—if $\bar{m}(t) < \theta$, DENV infection cannot be determined; log-normal$_\theta(\bar{m}(t)|\log m(t), \sigma_m)$ denotes a truncated log-normal probability density evaluated at $\bar{m}(t)$, which is centered on the true latent midgut viral quantity, $m(t|d)$, which is obtained by solving Eqs (1)–(3). We assume a log-normal distribution since the variation around the mean fits visibly increased with the virus levels. Note, the ODE-derived quantity is assumed to depend on $d$, since the initial quantity of virus taken into the lumen depends on the dilution of the blood meal. For $\bar{m}(t) > \theta$, we use a log-normal distribution that has been truncated to have a lower bound of $\theta$. There are two possible ways in which $\bar{m}(t) = 0$ can occur: either the midgut was not invaded, with probability $(1 - \phi(d))$; or there was successful invasion but the viral titer was below the detection limit, with probability $\phi(d)\text{F}(\theta|\log m(t|d), \sigma_m)$: here, $\text{F}(\theta|\log m(t|d), \sigma_m) = \mathbb{P}(\bar{m}(t) < \theta)$ is the cumulative distribution function of a log-normal distribution with location parameter $\log m(t|d)$ and scale parameter $\sigma_m$.

For the dichotomous presence data, we assume a binomial sampling distribution:

$$y(t|d) \sim \text{binomial}(n(t|d), p_m(t|d)), \tag{7}$$

where $0 \leq y(t|d) \leq n(t|d)$ denotes the count of mosquitoes with infected midguts out of $n(t|d)$ mosquitoes dissected at time $t$ with dilution $d$. The model for $p_m(t|d) := \mathbb{P}(\bar{m}(t) > \theta)$ mirrors the assumptions made to handle continuous data: for an infection to be recorded in the midgut, successful invasion must have occurred and the DENV titer must have exceeded the detection threshold:

$$p_m(t|d) = \phi(d)(1 - \mathrm{F}(\theta|\log m(t|d), \sigma_m)). \tag{8}$$

Note that Eq (8) combines both the logistic-type model given by Eq (5) for the presence/absence of infection and the ODE system solution to determine the probability that DENV titers exceed the threshold.

We now introduce the model for the viral quantity measured in the legs and body, $h(t)$. We again assume no detection of virus if the concentration is below $\theta$, and that it is possible that a midgut never becomes infected—preventing infection of subsequent tissues. We also allow for the possibility that a midgut may be infected but that escape may not occur due to mechanisms not included in our mechanistic system of ODEs. Specifically, we introduce a parameter: $0 \leq \xi \leq 1$, which represents the probability of escape from an infected midgut. The probability density of viral loads is then described by:

$$p(\bar{h}(t)|d) = \begin{cases} \phi(d)\xi\text{log-normal}_\theta(\bar{h}(t)|\log h(t|d), \sigma_h), & \text{if } \bar{h}(t) > \theta, \\ (1 - \phi(d)) + \phi(d)\xi\mathrm{F}(\theta|\log h(t|d), \sigma_h), & \text{if } \bar{h}(t) = 0, \\ 0, \ \text{otherwise}, \end{cases} \tag{9}$$

where, again, we assume a log-normal measurement noise model. For the dichotomous data, we assume a binomial sampling model,

$$y(t|d) \sim \text{binomial}(n(t|d), p_h(t|d)), \tag{10}$$

where $0 \leq y(t|d) \leq n(t|d)$ denotes the count of mosquitoes with infected legs out of $n(t|d)$ mosquitoes dissected at time $t$ with dilution $d$. The probability of these tissues testing positive, $0 \leq p_h(t|d) \leq 1$, is assumed to follow:

$$p_h(t|d) = \phi(d)\xi(1 - \mathrm{F}(\theta|\log h(t|d), \sigma_h)). \tag{11}$$

We performed experiments where mosquitoes were blood-fed at time $t = 0$ and the levels of CHP, which acts as a measure of basal lamina damage, were quantified over time (see *Basal lamina collagen IV damage* section). Subsequent to blood feeding, the level of CHP was assumed to follow:

$$\frac{dc(t)}{dt} = -\eta(c(t) - c^*), \tag{12}$$

where the same $\eta$ appears in Eqs (2) and (12) because, lacking a direct measure of basal lamina permeability, we assume that the rate of decline of CHP occurs in lockstep with changes in basal lamina recovery and permeability. We assume $c(0) = c_0$ is the initial CHP level. We assume that the measured CHP level, $\bar{c}(t)$, differs from the underlying level, $c(t)$ due to a measurement error model:

$$\bar{c}(t) \sim \text{log-normal}(\log c(t), \sigma_c), \tag{13}$$

where $\sigma_c > 0$ denotes the width of the measurement error. We assume a log-normal error model since it was visually discernible that there was greater variation in measured CHP when the mean levels were higher.

We tested our system of ODEs (Eqs (1) to (4)) for structural identifiability using SIAN (Structural Identifiability ANalyzer) [43], which we accessed via an online GUI [44]. This showed that all of the ODE model parameters were globally structurally identified, including the initial conditions.

**Priors.** We used a Bayesian modelling paradigm to fit the model to experimental data. Our chosen priors are shown in S1 Table. For some parameters, the data were used to construct priors since the scales of the data inevitably affected this choice: the two parameters affecting the dynamics of CHP, $c_0$ and $c^*$, were assigned (wide) priors with means close to the corresponding sampling means at $t = 0$ for blood-fed and unfed mosquitoes, respectively; the priors for the initial infection dose, and the midgut and leg infection carrying capacities were chosen to reflect that the data were scaled relative to the mean DENV titers in each tissue. The parameter $\eta$ effectively determines the rate at which the basal lamina layer recovers to pre-blood fed levels, and we chose a prior which gave the vast majority of probability mass to recovery within the first few days following blood feeding following experimental results [38]. The priors for the parameters of the generalised logistic function (Eq (5)) were chosen to allow a wide range of dose-response relationships (see S1(A) Fig).

In the Results, we highlight the sensitivities of the model results to both changes to the fitted parameter values and the priors themselves.

**Parameter values across experiments.** Since the initial viraemia taken in by the mosquitoes likely varies systematically across the different feeds, we assume that mosquitoes fed with a dilution of 1:1 have an initial viral load of $l_0$; and those with dilutions of 1:$d$ have an initial viral load of $l_0/d$. There were systematic differences in prevalence between the 1:5 dilution experiments for the dichotomous measurements and the continuous measurements. To account for this, we introduce an additional experiment-level parameter for the dichotomous experiments, $\zeta > 0$, meaning that the initial viral load is $\zeta l_0/d$. In what follows, we estimate $\zeta$.

## Numerical solution and model fitting

Our system of ODEs is stiff and was numerically integrated using Stan's backwards differentiation formula method using the default error tolerances [45].

The qPCR data were scaled by normalising them with the tissue-specific means so that the data were nearer the unit scale, which better facilitated computational sampling. To estimate the model posteriors, we performed Markov chain Monte Carlo (MCMC) sampling using Stan's default sampling method [45], which is based on Hamiltonian Monte Carlo [46]. The sampling was conducted using 4 Markov chains, each run for 1200 iterations (with the first half of the iterations discarded as warm-up), and convergence of the Markov chains was diagnosed by $\hat{R} < 1.01$ and both bulk- and tail-effective sample size (ESS) exceeding 400 [47].

All experimental data and code (R & Stan) for running our analyses are openly available in a Github repository and associated Zenodo repository (see *PLOS paper* tagged release in github.com/ben18785/within_mosquito_arbovirus_paper; and [48]). The repository uses an R-specific pipeline tool, *targets* [49], which facilitates reproducibility by allowing users to straightforwardly rerun elements of the analyses.

The posterior estimates of the parameters, including summary and convergence measures are shown in S4 Table. To illustrate the identifiability of the model's parameters, in S2 Fig, we show the pairwise posterior draws for those 7 parameters which had the highest mean pairwise correlations with those from other parameters.

## Logistic regression analysis

We performed a logistic regression analysis to test whether changes in infection dose affected dissemination rates. The regression, including all terms, was of the form:

$$y_i \sim \text{binomial}(n_i, p_i), \quad p_i := \Lambda(\alpha + \beta_1 t_i + \beta_2 e_i + \beta_3 d_i), \quad (14)$$

where $y_i$ denotes the number of positive legs out of a total number of dissected specimens $n_i$ for experimental setup $i$: $t_i$ denotes DPI for that setup; $e_i$ denotes whether the data were from the continuous or dichotomous experiments; and $d_i$ denotes the dilution of the infection dose; $\Lambda(x) = 1/(1 + \exp(-x))$ is the logistic function.

To test the predictive value of including dilution in the regression, we fitted an additional regression model where this term was neglected (i.e. $\beta_3 = 0$ in Eq (14)). We then performed a likelihood ratio test comparing the two models.

## Results

### Midgut infection occurs rapidly following an infectious blood meal and the chance of successful infection is strongly dose-dependent

It is thought, that in order for an infection to occur, virus must invade the mosquito midgut epithelium prior to formation of the peritrophic matrix, which previous work suggests starts to form 4–8h post-feeding [50]. Our experimental data and modelling support this early viral invasion hypothesis. While the first time point for our dissections was 3 DPI, these and subsequent dissections indicate that there were minimal changes in the proportion of midguts infected over time, particularly for higher infection doses, suggesting that infection occurs prior to this timepoint (points in Fig 2A with comprehensive data shown in S3 Fig). Our model simulations are consistent with the experimental data and indicate that most midguts that become infected do so in the first day post-feeding (lines in Fig 2A) [51]. While midgut infection with Zika virus has been seen 24 hours post-infectious blood meal, future experiments where midgut infection data are collected at earlier timepoints for additional viruses are necessary to confirm the timing of midgut infection [51].

Our results also indicate that, if more virus is ingested in the blood meal, there is a higher probability that the virus successfully establishes in the midgut (Fig 2A points and lines). These findings are supported by the results of another modelling study that used a stochastic model of infection dynamics [29]. Our model simulations indicate that the highest infection doses (purple line in Fig 2A) establish in the first few hours post-infection whereas for lower doses, the process of invasion or time to viral detection may take longer (yellow line in Fig 2A). This finding is also supported by Fig 3 which shows that the DENV titers for the lowest concentration bloodmeal are low 3 DPI, which may introduce a lag between infection occurring and its subsequent detection. In Fig 2B, we consider 5 DPI and plot the modelled dose-response relationship (black line) and overlay the experimental data (black points). This suggests that there is a strongly non-linear effect of viral dosage on probability of successful establishment, with doses below a critical threshold unlikely to result in successful infection. Human DENV-2 titers vary considerably between individuals ($\sim$2–11 log10 RNA copies/ml) and thus the viral dose ingested by mosquitoes differs substantially between hosts [18, 52]. A concentration of 6.29–6.51 log10 RNA copies/ml of DENV-2 was found previously to result in 50% of the fed mosquitoes becoming infected when fed on DENV-2 patients [18, 52]. In our assays, an infection rate of 40–60% was achieved at a final concentration of 8.11–8.36 log10 FFU/ml (1:12 dilution). This supports previous observations that similar titers of virus in

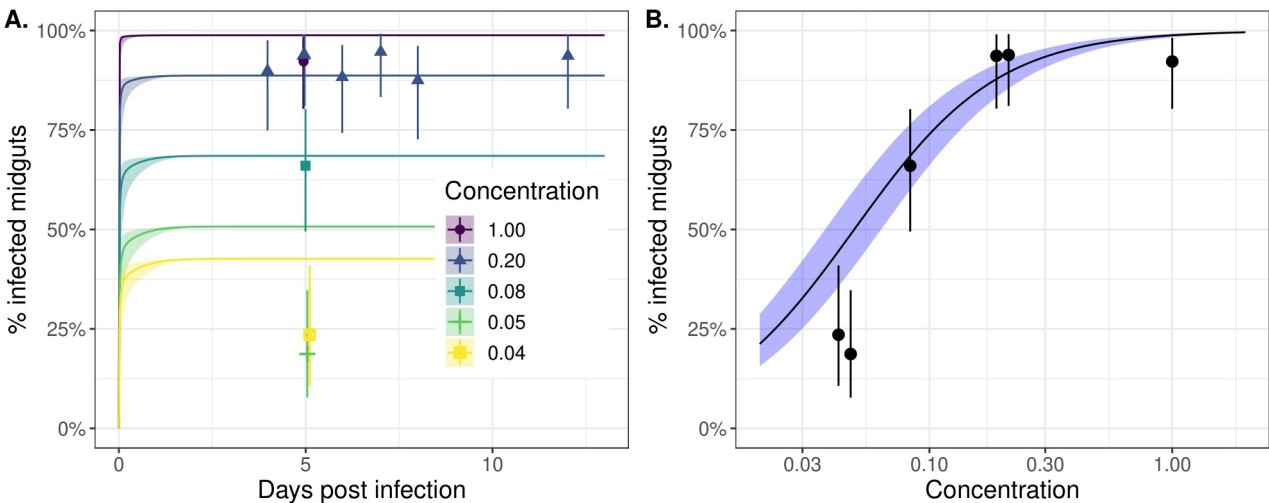

**Fig 2. Midgut infection occurs rapidly following an infectious blood meal and the likelihood of successful infection is strongly dose-dependent.**
Panel A shows the proportion of positive midguts as a function of days post-infected blood meal (DPI) for a range of infectious blood meal doses (coloured lines) relative to our highest "base" dose (concentration = 1; purple line). The points represent the experimental data, and the solid lines correspond to model simulations. Panel B shows the proportion of positive midguts on day 5 post-infectious blood meal as a function of infected blood meal dose relative to the base. Both plots show results only for the dichotomous feeding experiments. The points and whiskers indicate the experimental data, and the line and shaded ribbons represent the model simulation. The solid points indicate the posterior median proportions, and the whiskers indicate the 2.5%-97.5% posterior quantiles, assuming a uniform prior on the proportion. The lines represent the posterior median simulation and the ribbon represents the 2.5%-97.5% posterior quantiles in the median. In both plots, the modelled % infected midguts values were determined by calculating the probability the titer exceeds the detection threshold, specified by Eq (8).

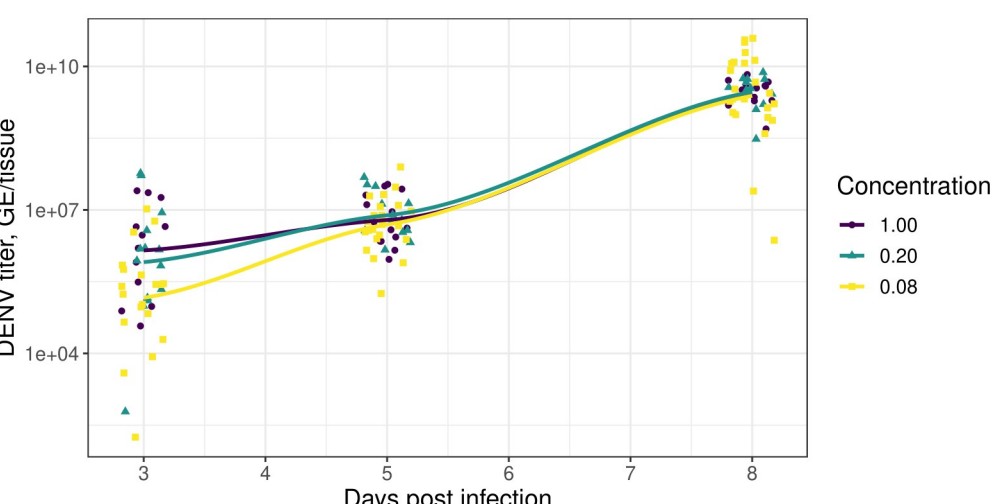

**Fig 3. Blood meal dose affects initial midgut viral titers but not the ultimate viral levels.** The points indicate the raw experimental measurements of virus titer. The lines represent locally estimated scatterplot smoothing (LOESS) curves fit to the data using the default *ggplot2* specifications [54]. The colouring corresponds to the concentration of the infected blood meal relative to the highest dose used. The points have horizontal jitter added to them.

artificial feeding settings result in lower mosquito infection rates when compared to mosquitoes fed on infected animals [53].

## Blood meal dose affects the midgut viral titer shortly after invasion but not the ultimate level

In Fig 3, we show the time course of raw experimental measurements of DENV titer (points) across three different experimental concentrations for those mosquitoes that became infected; overlaid on this, we show smoothed regression lines. These data illustrate that, subsequent to blood feeding, the level of infection in the midgut layer increases dramatically (the vertical axis is on a log-scale). At 3 DPI, mosquitoes feeding on the higher viral dilutions showed higher midgut DENV titers than the 1:12 group (Kruskal-Wallis rank sum test performed since data were non-normally distributed: 1:1 versus 1:12, $\chi^2$ = 12.8, df = 1, $p < 0.01$; and 1:5 versus 1:12, $\chi^2$ = 8.9, df = 1, $p < 0.01$; the means for the 1:1 versus 1:5 titers were not significantly different) but this early lead is transient, and the viral titer in the midgut layer does not strongly depend on the initial blood meal dose by 8 DPI. Due to this, we assume that the carrying capacity of the midgut layer, $\kappa_m$, is independent of infection dose (see Eq (3)).

## Blood meal dose affects the timing of disseminated infection

Our model also predicts that higher initial infection doses result in faster viral growth in the midgut infection titer (Fig 4: top panels, blue lines and ribbons). As the basal lamina layer is thought to be partially permeable to virus particles shortly following feeding and may be permeable even at baseline, higher numbers of viral particles may lead to more encounters with areas where the basal lamina is compromised and, therefore, more opportunities for a speedy escape from the midgut [9]—a hypothesis embodied in our model (see Methods; Fig 5A) and supported by our experimental data.

In Fig 5B, we plot the percentage of virus-positive legs (points), indicative of viral dissemination, in those specimens which had infected midguts, and we colour these according to the concentration of the initial dose. A logistic model regression analysis using these raw data (see Methods) found that changes in initial blood meal dilution affected the leg positivity (likelihood ratio test: $\chi_1^2 = 4.43, p < 0.05$) and that increases in dilution coincided with reductions in positivity (S3 Table). The faster rates of dissemination predicted by our model also result in faster growth in the leg infection titer for higher doses, as predicted in Fig 4 (bottom row, blue lines and ribbons).

Faster growth in viral titer in the legs means that disseminated infection is detected earlier in a higher proportion of specimens (see Fig 5B). A consequence of this is that, on average, less time is required for detectable disseminated infection to occur for higher initial doses (Fig 5C). However, both the low and high ends of this plot involve extrapolating the model beyond the range of the data (i.e. outside the dashed lines in Fig 5C). Given that natural DENV-2 titers in human patients extend beyond the titers tested in this study (S2 Table), it is unclear how this dose-dependency will hold in the wild—a point to which we return in the discussion [18, 52]

## Midgut invasion and dissemination rates are sensitive to fluctuations in parameters representing viral establishment and less sensitive to changes in the rate of viral digestion in the lumen

Whilst we developed our model by fitting to data using DENV, the biological mechanisms it encodes are qualitatively similar across other arboviruses, but the rates of the various processes are likely to differ. Mutations in the CHIKV envelope protein have been shown to increase

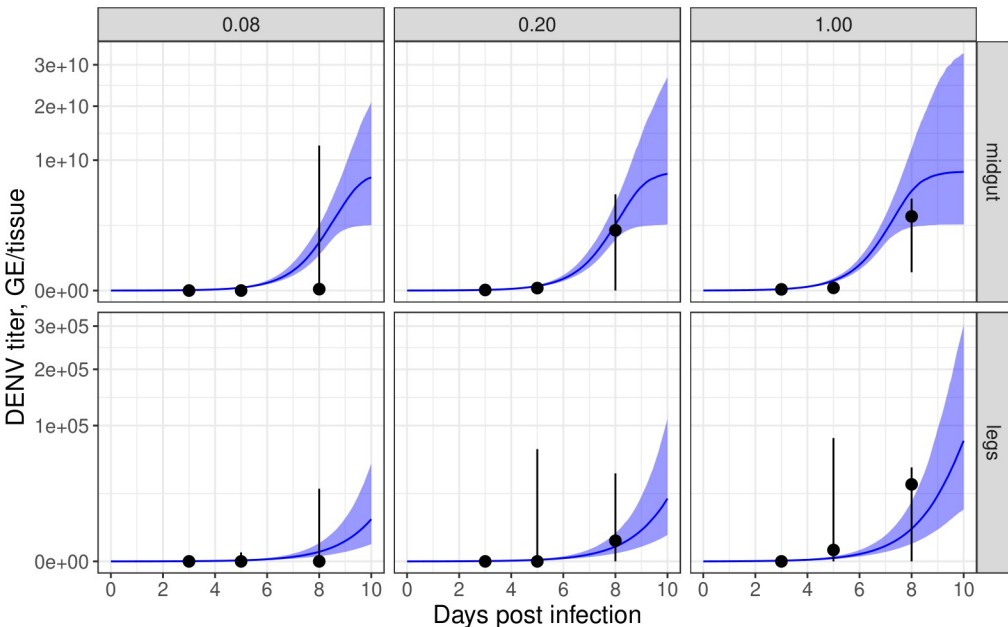

**Fig 4. Blood meal dose affects the time-profile of midgut and leg infection titers.** The top / bottom rows correspond to midgut / leg infection; the panels correspond to relative initial blood meal dose. The blue lines and ribbons indicate modelled results (lines correspond to posterior predictive median simulations of the modelled mean; ribbons to 2.5%-97.5% posterior quantiles in the modelled mean). The black points and whiskers indicate the raw experimental data (points correspond to medians; whiskers to 10%-90% quantiles).

midgut infectivity and rates of dissemination in *Aedes albopictus* mosquitoes and have been implicated as a cause of outbreaks of chikungunya virus in Reunion island in 2005–2006 [55] and later in Southern India [56]. Changes in temperature have also been shown to affect the EIP for a number of mosquito-borne viruses [57–59], and it is possible that midgut infection and dissemination rates are perturbed by such environmental fluctuations.

We now use our mathematical model to explore how changes to three parameters which characterize the ability of an arbovirus to invade the midgut affect midgut infection and dissemination. We focus on: $\gamma$, which controls the rate at which the virus is digested in the lumen; $a$ and $k_{lm}$ which, in combination, control the rate of invasion. We vary each of these parameters in isolation, holding the others at their posterior mean estimates. We then compare the rates of midgut invasion and presence of infection in the legs (i.e. disseminated infection) at the posterior mean parameter values to those obtained considering hypothetical multiples of each of the parameter values across two orders of magnitude.

In S4 Fig, we plot the midgut (top row) and leg (bottom row) infection profiles as each of $\gamma$, $a$ and $k_{lm}$ are varied (one parameter per column). The dashed lines show the infection profiles at the posterior mean estimates. These simulations show that a two-order magnitude change in the rate of virus degradation modestly affects the rate of midgut and leg infection, with slower rates of degradation leading to faster infection. Perturbations to the saturation parameter, $a$, and the midgut invasion rate parameter, $k_{lm}$, generally constitute larger changes, particularly in the rate of dissemination. Decreases in $a$ typically lead to a higher rate of midgut invasion for a given infection titer, leading to a faster rate of infection; and increases in $k_{lm}$ monotonically increase the rate of invasion, leading to more rapid infection.

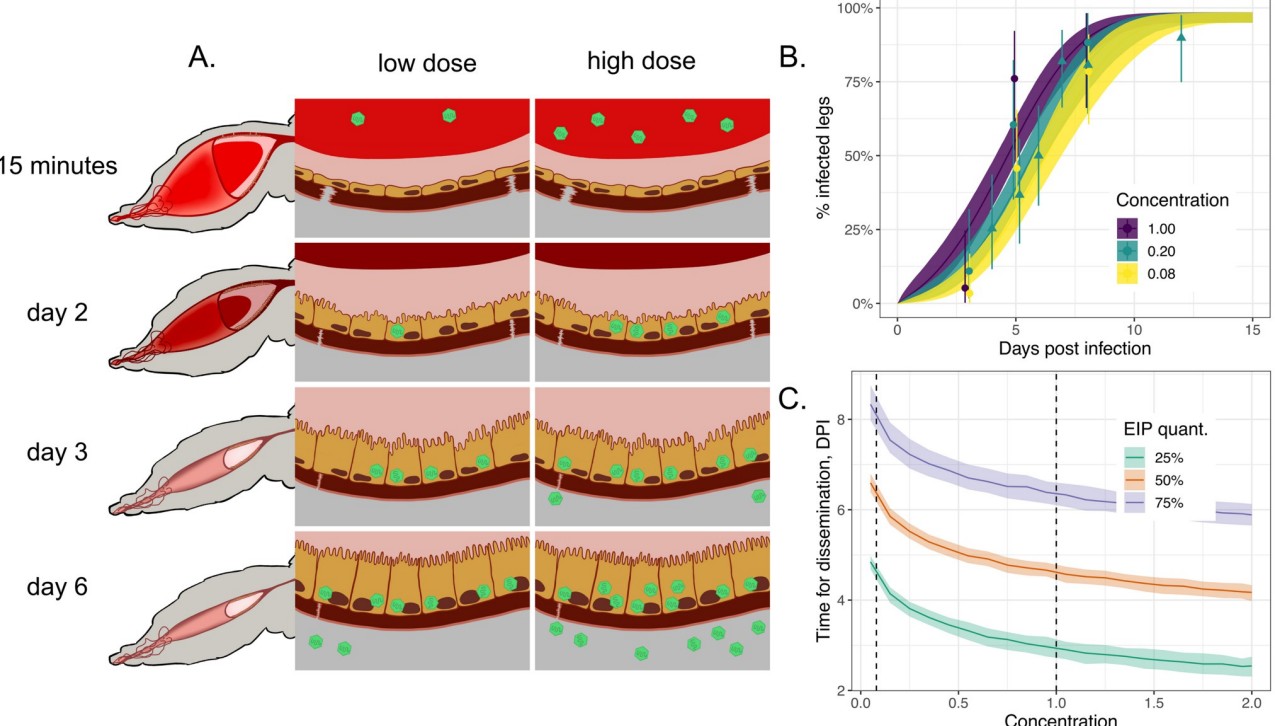

**Fig 5. Blood meal dose affects the timing of dissemination.** Panel A shows a schematic of two mosquito midguts over the initial course of infection (virus particles shown in green): the left-hand sequence of images corresponds to a mosquito fed on a low dose of virus; the right-hand sequence, to a mosquito fed on a high dose. Panel B shows the positivity of legs conditional on successful midgut infection as a function of concentration (colouring). Points / triangles show the posterior medians for each experimental data point for the continuous / dichotomous experiments and the ranges show 2.5%-97.5% posterior quantiles assuming a beta prior on positivity. Panel C shows how the modelled time taken to reach 25%, 50% and 75% disseminated infection is affected by blood meal dose; the vertical dashed lines delimit the range of experimental infection doses explored (i.e. anything outside the range is extrapolative). In both panels, the lines show posterior median model simulations, and the shading shows the posterior 2.5%-97.5% posterior modelled quantiles. In panel B, the modelled % infected legs values were determined by calculating the probability the titer exceeds the detection threshold, specified by Eq (11).

Taken together, these results indicate the sensitivity of midgut and disseminated infection to perturbations in parameters affecting the virus' ability to invade the midgut.

### Transient basal lamina permeability following a blood meal can explain earlier dissemination in mosquitoes taking successive blood meals

Previously the basal lamina layer which surrounds the midgut epithelium was thought to be impermeable to arboviruses because its pore sizes are smaller than the diameter of virus particles when undamaged [38, 60]. Following blood feeding, however, the basal lamina layer becomes damaged and appears to be transiently permeable to virus particles [9, 38, 51], which has been proposed as a mechanism explaining earlier dissemination in mosquitoes acquiring sequential blood meals (see Fig 6A).

In Fig 6B, we present experimental data that quantify the degree of damage to this layer, as measured using a peptide called CHP (see Methods) that binds to damaged collagen IV—a main component of the basal lamina. This plot shows that collagen IV damage is highest shortly following blood feeding but that damage and basal lamina integrity recover over time and return to a baseline comparable to that in unfed mosquitoes by four days after blood feeding.

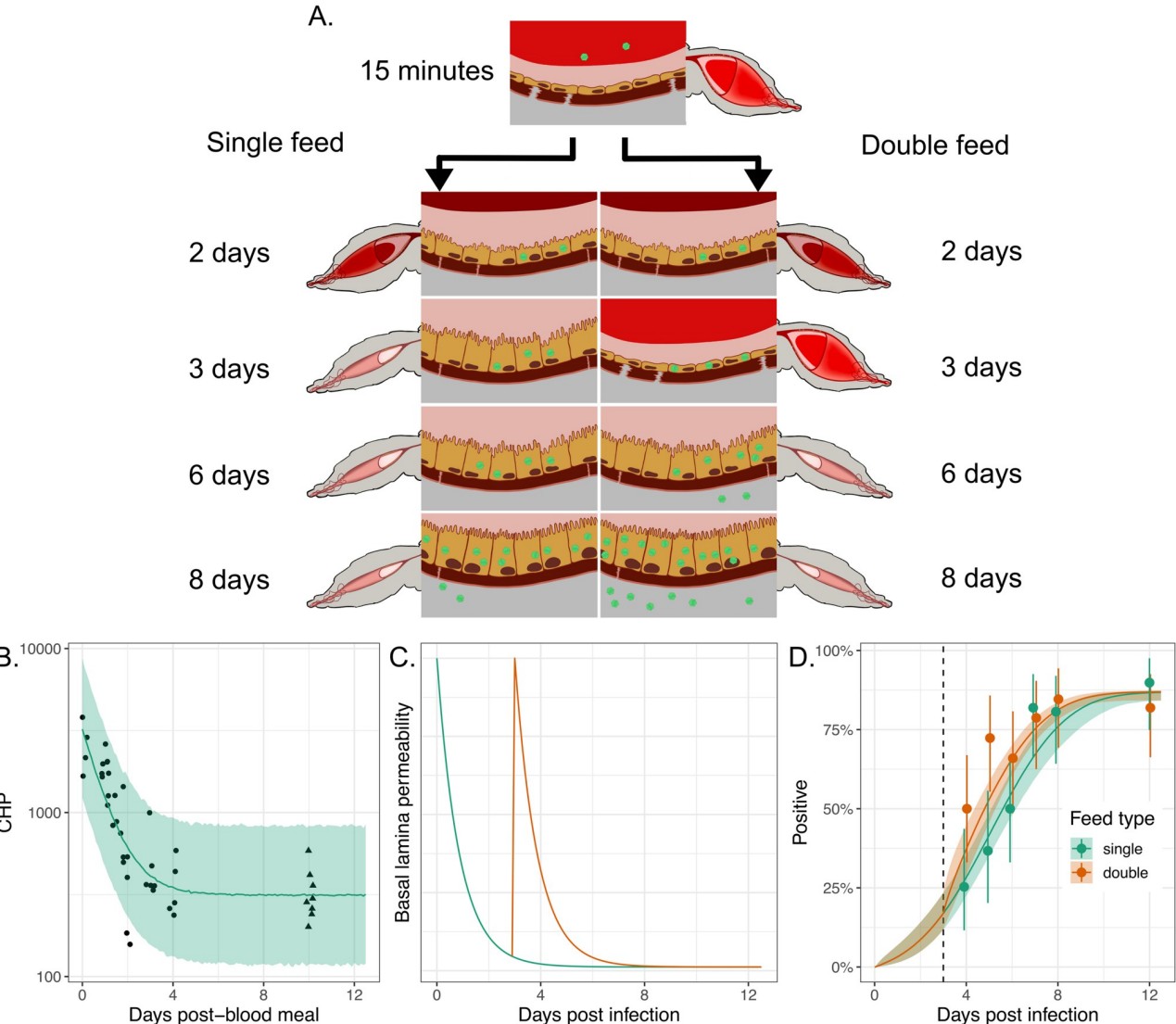

**Fig 6. Transient basal lamina permeability following a blood meal can explain earlier dissemination in mosquitoes taking successive blood meals.**
Panel A shows a schematic of viral invasion of the midgut epithelium layer with virus particles shown in green. The left-hand sequence of images corresponds to a mosquito bloodfed once, on day 0. The right-hand sequences corresponds to a mosquito fed once on day 0 and again on 3 days post the initial bloodfeed; this second feed leads to tears in the basal lamina leading to more rapid viral escape from the midgut layer. Panel B shows data (points and triangles) from experiments where CHP, a measure of damage in the basal lamina layer, was measured post a blood feed at time 0 (see Methods). The points represent observations taken at the times indicated; the triangles represent observations taken on unfed mosquitoes, here assumed to proxy for the level of CHP 10 days post-blood meal (DPBM). The green line and uncertainty ribbons (2.5%-97.5% posterior quantiles) show the model fits to the data. Panel C shows the modelled basal lamina integrity (measured by $x(t)$ in Eq (2)) for single- (green line) or double-fed (orange line) mosquitoes, where the second feed occurs 3 DPI. These lines represent posterior median simulations. Panel D shows the proportion of infected legs indicating dissemination infection for either single- or double-fed mosquitoes, where the second feed occurs 3 DPI (dashed line). The points show the experimental data, and the solid lines (posterior medians) and ribbons (2.5%-97.5% posterior quantiles) show model simulations. The solid points indicate the posterior median proportions, and the whiskers indicate the 2.5%-97.5% posterior quantiles, assuming a uniform prior on the proportion. In panel D, the modelled % infected legs values were determined by calculating the probability the titer exceeds the detection threshold, specified by Eq (8).

By combining our mathematical model with the CHP data, it is possible to quantify the timescales of basal lamina repair for a mosquito that feeds only once (solid line in Fig 6B). We assume that the dynamics of basal lamina damage mirror the permeability of this layer to virus particles subsequent to blood feeding. As a result, mosquitoes which feed again provide an additional opportunity for the virus to escape the midgut (Fig 6C).

In Fig 6D, we show new experimental data for mosquitoes which are fed a DENV-2 spiked blood meal on day 0 where a subset of mosquitoes are fed again 3 DPI with a noninfectious blood meal. This figure shows the proportion of mosquitoes with infection detected in the legs and indicates that mosquitoes fed an additional blood meal have earlier dissemination, results consistent with those presented previously [9]. It has been proposed that, if given a second, noninfectious blood meal several days after an infectious feed, virus particles that invaded the midgut epithelium during the first feed have been given time to multiply and are now present in high quantities and in close proximity to the basal lamina. When an additional blood meal occurs, basal lamina damage again spikes and these particles have a greater chance to encounter a damaged area and to escape into the haemolymph—hastening the course of infection [9]. If not given a second blood meal, viral particles present in the midgut epithelium experience a less permeable basal lamina layer which inhibits their ability to escape from the midgut. Our model shows that this mechanism can explain why double-fed mosquitoes have earlier disseminated infection than those receiving a single blood meal (solid lines in Fig 6D).

Previous experimental studies have demonstrated an asymmetry with double feeding: if a mosquito is fed an infected blood meal (at time 0) followed by uninfected blood meal (at a later time), the course of infection is hastened and dissemination occurs earlier. If instead the mosquito is fed an uninfected blood meal followed by an infected one, then the time between ingestion and dissemination is indistinguishable from that in mosquitoes only fed once with infectious blood [9]. This reflects basal lamina repair dynamics and emphasizes that damage to the midgut basal lamina is transient. If mosquitoes are infected via the first blood meal and virus is able to invade the midgut cells and multiply, by the time a second blood meal occurs, 3–4 days later, there are many virions present in the midgut epithelial layer that can exploit the basal lamina damage associated with blood feeding, leading to faster dissemination [9]. If mosquitoes are fed an uninfected blood meal followed by an infectious feed, they do not have the bank of virions within the midgut layer when damage from the second blood meal occurs and, therefore, the dissemination dynamics resemble that in mosquitoes fed a single infectious feed. In S5 Fig, we show that our mathematical model recapitulates this trend seen in the experimental data.

## The double blood feeding effect is sensitive to parameters which affect viral proliferation and escape of the virus from the midgut

We have demonstrated that transient lamina permeability allows a secondary noninfectious blood feed to hasten the course of disseminated infection. We now investigate how such a "double feed effect" is affected by changes in the rate at which the virus proliferates in the midgut, $\alpha_m$ and two parameters, $\eta$ and $k_{mh}$, which, in combination, affect the rate at which the virus escapes the midgut. Higher values of $\eta$ result in faster recovery of the basal lamina layer, and a shorter time window during which viral escape from the midgut is possible; higher $k_{mh}$ results in higher rates of escape from the midgut, although these effects are dynamically moderated by basal lamina permeability.

In S6 Fig, we use our model to show how changes in each parameter affect the rate of dissemination in mosquitoes that were singly fed and those fed an infectious blood meal at 0 DPI followed by a second noninfectious blood meal at 3 DPI. The top row of S6 Fig shows that if

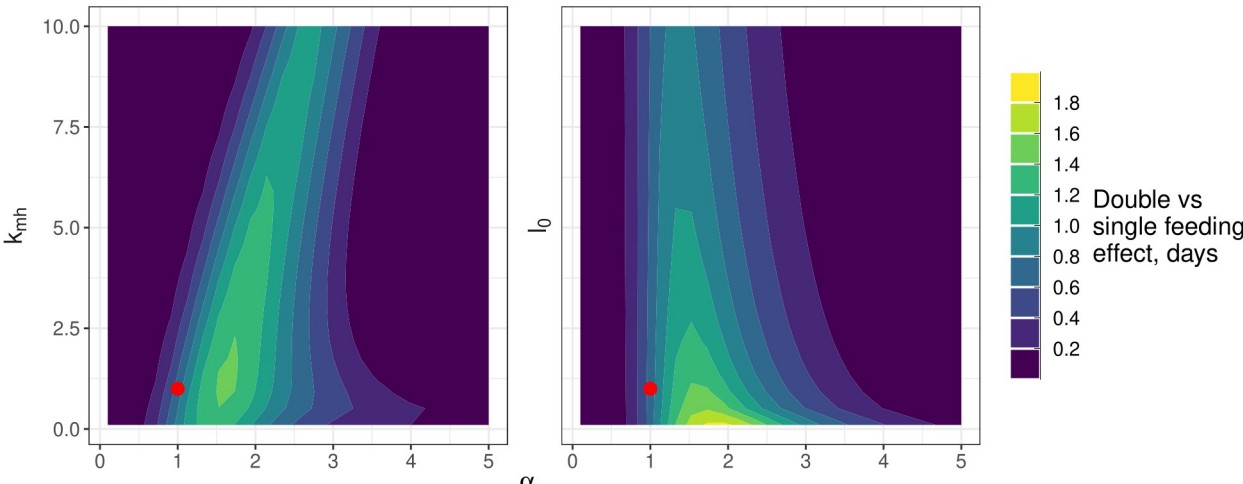

**Fig 7. The dependence of the double feed effect on the midgut viral proliferation rate ($\alpha_m$), the rate of escape from the midgut ($k_{mh}$) and the initial bloodmeal titer ($l_0$).** The colouring indicates the difference in the time taken to reach a midpoint in viral dissemination for double versus single feeding: lighter colours indicate a faster rate of dissemination for double feeding. The axes show the multiples of the posterior mean estimates of each of the parameters used for the simulations. The red point indicates the double feeding effect estimated at the posterior mean parameter values. Note, that the simulations were performed and gridded values of each parameter and the resultant heatmap has been smoothed using the default options of the `geom_contour_filled` option in the *ggplot2* package [54].

the rate of viral proliferation in the midgut is either one-tenth or ten times the model-fitted result, then there would be minimal differences between mosquitoes fed a single blood meal and those fed two sequential blood meals: at lower rates, the virus replicates too slowly to take advantage of the time window for escape provided by the second blood feed; at higher rates, the virus is able to escape within the window provided by the first blood meal and a secondary blood meal provides only a slight boost. The middle row of S6 Fig shows that, as the rate of basal lamina recovery increases, the greater the effect of double feeding. A faster rate of basal lamina recovery results in shorter time windows for viral escape, meaning a secondary blood feed—providing a secondary window of escape—leads to substantial increases in the rate of viral escape. The bottom row of S6 Fig shows that, if the rate of viral escape from the midgut increases, the double feeding effect diminishes. This is because at faster rates of viral escape, the time window of escape provided by the first feed is sufficient to allow substantial dissemination. Our results thus point towards a "sweet spot" in parameter values corresponding to a substantive double feeding effect, which we now investigate further.

We can quantify a "double feed effect" as the difference in time taken for a mosquito fed a single blood meal versus a mosquito fed two sequential blood meals to reach 50% of its maximum level of disseminated infection levels. In the left-hand panel of Fig 7, we plot this "double feed" effect as a function of simultaneous changes to $\alpha_m$ and $k_{lm}$. This further study highlights the narrow window of possible $\alpha_m$ values where there is a double feeding effect and a relative insensitivity to changes in $k_{lm}$. It also indicates that double feed effects that are markedly larger than those we estimated are possible with small changes to $\alpha_m$. In the right-hand panel of Fig 7, we explore how the double feed effect may respond to changes in the bloodmeal dose. This suggests that high bloodmeal doses can effectively saturate the system with virus, leading to fast dissemination and a minimal additional effect of a second bloodmeal on dissemination rates.

## Discussion

Understanding the biological mechanisms which influence the life cycle of viruses within their mosquito vectors is crucial for developing public health interventions which aim to target these life stages. In this study, we presented detailed data from laboratory experiments across multiple infection doses and multiple time points, which were then used to develop a mathematical model of viral population dynamics within mosquitoes. The model allows us to quantify how the infection dose affects midgut infection and the rate of dissemination; it has also enabled us to explain why mosquitoes receiving multiple blood meals may have faster dissemination. Our work also hints that the magnitude of this double-feeding effect may be particularly sensitive to changes in the rate of viral proliferation in the midgut, which suggests the effect size may differ across arbovirus-mosquito combinations. Our model represents a first step towards what we envisage as a series of mechanistic models at different scales—from the subcellular to the tissue level—which can be used as tools for understanding this complex system.

The mathematical model we developed comprises a system of autonomous nonlinear ODEs, which implicitly assumes that the number of individual virus particles within each mosquito tissue is large. This assumption is valid for each of the tissues we modelled at later times during their infection but fails to account for known bottlenecks where the number of virus particles are thought to be relatively few. Despite these limitations, our model closely matched the gathered experimental data taken from 3 days post-infection onwards. In future work, we plan to probe the initial stages of midgut infection by collecting experimental data from early on during the midgut infection in order to verify our modelled results in this period.

A small number of midgut epithelial cells are thought to become infected by direct invasion of virus from the lumen, possibly due to relatively few virions contacting the right type of midgut epithelial cell [61, 62]. Invasion of the midgut is thought to occur rapidly following blood feeding as otherwise the virus particles are lost through diuresis or become trapped by the peritrophic matrix, a chitinous sac secreted by the midgut epithelium into the gut lumen during blood digestion [14]. Whether wild populations avoid these bottlenecks is unclear since infectious doses in the wild are thought to be considerably lower than those required in the laboratory to produce successful infection [53, 63], and midgut infection has been demonstrated to present a particularly strong barrier at low infection doses [64]. Some have even suggested that these bottlenecks may be evolutionarily advantageous by selecting against defective virus particles early on during infection [29]. At a given titer, however, asymptomatic patients yielded a higher proportion of infected mosquitoes and higher mosquito viral titers; indicating that while DENV titer is important, other factors such as host immune responses may play a role in determining vector infection [65]. The mathematical model we developed does not allow individual-level interaction of the virus with the mosquito tissues (and the stochasticity this brings), so does not allow mosquitoes to clear infection. To account for this known characteristic, we used a phenomenological statistical model to allow specimens to remain uninfected. Others have shown that stochastic individual-based models of the invasion process can be used to understand the dose-dependency in the probability a mosquito becomes infected [29]; interestingly, this other work also hints at there being a strong dose-dependency in dissemination.

The midgut escape mechanism has been shown to present a major bottleneck in viral population diversity for a range of arboviruses [66–68]. Here, we represented the temporary window through which escape is possible by allowing the basal lamina layer to be partially permeable to virus particles only in the first few days following blood feeding. This is likely an oversimplification as the fraction of mosquitoes with disseminated infection continues to rise

even after the blood meal is digested and the basal lamina layer is repaired, indicating that the basal lamina may not be completely impermeable even at baseline or that disseminated viral levels may initially be lower than the limit of detection [38, 51]. As only a handful of virus particles are thought to be responsible for disseminated infection, it seems plausible that some mosquitoes may fail to become infectious because no virus particles manage to escape the midgut; our continuous model does not allow such a possibility [64]. Extending either our framework to model individual virus particles or extending other stochastic models [29] to incorporate interactions with the basal lamina layer could help to understand whether purely random phenomena can explain why some mosquitoes become infectious and others do not under the same experimental conditions or whether other mechanisms, such as individual differences in the initial blood meal size, virus inoculum, or variation in immune response account for this difference.

The model we presented here was spatially averaged within each tissue, and we considered spatial spread only at a gross level: modelling the movement of virus particles between tissues as transitions between model compartments. It is not clear what importance, if any, the processes by which the virus infects and then spreads between neighbouring midgut cells have on the course of viral infection of mosquitoes. The initial site of infection in the midgut epithelium has been shown to be localised primarily in the posterior midgut [61], and it is possible that cell-to-cell viral spread and proliferation allows the virus to find the relatively few sites of basal lamina weakness following blood feeding [9]. A spatial model which accurately modelled the mechanics of the basal lamina layer deformations could help to probe this hypothesis.

Given the heterogeneous outcomes of infection experiments, it is reasonable to ask whether there are heterogeneous inputs (i.e. blood meal or infection doses) or processes (e.g. variation in mosquito immune responses) that explain this. Mosquito dissection experiments in general are problematic for examining individual dynamics in pathogen dynamics—a mosquito can only be dissected once, providing only a snapshot of the processes involved. This means that experiments, such as ours, dissect many mosquitoes at different time points, and the implicit assumption when interpreting these data is that the differences in dynamics between mosquitoes are sufficiently low that the overall dynamics can be pieced together from the snapshots. We followed such an approach here when fitting our model since we assumed a single set of parameters across all specimens, including those representing blood meal size, dose of infection and those capturing the mechanisms of viral spread and the mosquito response. Whilst there exist approaches for inferring subpopulation population dynamics from such snapshot data [69–72], it is unclear whether those methods could identify such different dynamics owing to the (likely) high measurement noise and relatively few mosquitoes dissected (relative to experiments where these techniques are more typically employed, like for cell cytometry where typically many thousands of cells are analysed). Alternative approaches which allow for a measure of individual-mosquito level infection dynamics by monitoring the level of virus in mosquito excreta or saliva that is expectorated during bloodfeeding show promise and further work is necessary to compare these techniques with the results obtained by dissection [40–42, 73].

While the titers of DENV-2 used in this study overlap with titers seen in dengue patients, we saw slightly lower rates of infection than expected at our lower dilutions of virus:blood (1:20 and 1:25) when comparing the titers we used (7.48–7.57 log10 FFU/ml) to those in dengue infected patients used to feed mosquitoes [18, 52]. While this difference was unexpected, dengue infectivity is highly variable even when mosquitoes are fed on infected patients and there is some evidence to suggest that feeding mosquitoes via artificial means instead of on an infected host can result in lower infection rates for the same titer of virus [18, 52, 53]. Further, infectivity can vary between viral strain even within the same dengue serotype [74]. Artificial

blood feeds use defibrinated blood that has been treated with anti-coagulant chemicals and may not capture the impact of host immune responses on subsequent mosquito infection. These factors may impact the rate of infection and may explain why we saw lower levels of infection in our experiments than when mosquitoes were fed on dengue patients [18, 52]. While we used higher DENV-2 titers that resulted in lower levels of infection, our DENV-2 titers are internally consistent across replicates and infection levels decrease with decreasing viral dose. Despite these differences, artificial feeds provide important insights into mosquito vector competence and within-mosquito viral dynamics and are still considered a standard method in vector biology. Although more work is required to better understand how to extrapolate from laboratory experiments to wild transmission, mathematical models may prove useful tools in this investigation.

Vector biologists have made great leaps in our understanding of the predominant mechanisms underpinning the life stages of pathogens within mosquitoes. However, due to the complexity of the vector-pathogen system and the interaction between many different cellular and subcellular elements, unpacking the impact of different processes often requires quantitative models. Thus, we argue, the process of learning vector and pathogen biology can be greatly enhanced through collaborations between biologists and mathematicians.

## Supporting information

**S1 Table. Summary of priors.** Here, $\text{dist}_a^b()$ indicates a distribution truncated so its domain is between $a$ and $b$; $\text{dist}^+()$ denotes a distribution truncated to have support only over positive values.
(PNG)

**S2 Table. Viral titers of virus:blood stocks fed to mosquitoes.** Date of experimental feeds and dilution of virus:blood fed to mosquitoes. Viral titers were measured using focus forming assay and viral foci indicating viable viral particles were counted and used to calculate focus forming units per milliliter (FFU/ml). Data type for each experiment indicates whether infection was measured in a dichotomous, positive:negative manner, or whether viral titers were assessed continuously via qPCR.
(PNG)

**S3 Table. Logistic regression results.** In both regressions, a binomial sampling model was assumed with a logistic link function. The logistic model success variable was the number of positive legs in those specimens with infected midguts. Note, *p<0.1; **p<0.05; ***p<0.01.
(PNG)

**S4 Table. Posterior estimates and diagnostic statistics.** The metric $d^*$ measures the discrepancy between the (marginal) posterior distribution of a parameter and its prior: a value of 1 indicates that the two distributions have no overlap; a value of 0 indicates that the two distributions are the same. $d^*$ is calculated by fitting unidimensional kernel density estimators to each of the two distributions and using these as classifiers (where the class with the highest probability density is the class prediction). $d^*$ is computed as $2(A - 0.5)$, where $A$ is the classification accuracy across an independent test set. The MCMC convergence diagnostics, $\hat{R}$ and the two bulk ESSs, are calculated using the *posterior* R package [75].
(PNG)

**S1 Fig. Prior predictive distributions.** Panel A shows summaries of the prior predictive distribution for $\phi(.)$ in Eq (5). In panels B and C, we show summaries of the prior predictive distribution for the logistic equation components of Eqs (3) & (4), respectively. Specifically, in these

panels, we show the solution of the logistic equation: $dy/dt = \alpha y(1 - y/\kappa)$ with $y(0) = 0.0001$ assuming the priors for the midgut (A) and legs (B). In all panels, the summaries were created using 1000 draws from the prior predictive distribution, and the ribbons show the 2.5%-97.5% quantiles and the black lines show the prior median. The orange lines show the posterior median estimates of each quantity.
(TIFF)

**S2 Fig. Posterior estimate correlations.** This plot shows pairwise posterior plots for draws of the parameter estimates for the seven parameters with the highest average pairwise correlations. In the plot, we indicate the significance level of the correlation estimates by asterisks: *p<0.1; **p<0.05; ***p<0.01.
(TIFF)

**S3 Fig. Extended midgut infection results.** Both panels show the proportion of positive midguts as a function of days post-infected blood meal (DPI) for a range of infectious blood meal doses (coloured lines) relative to our highest "base" dose (concentration = 1; red colouring). The points show the experimental data, and the solid lines show model simulations. The left-hand panel is an exact replicate of Fig 2A and corresponds to the dichotomous experiments; the right-hand panel corresponds to the continuous experiments. The points and whiskers indicate the experimental data, and the line and shaded ribbons represent the model simulation. The solid points indicate the posterior median proportions, and the whiskers indicate the 2.5%-97.5% posterior quantiles, assuming a uniform prior on the proportion. The lines represent the posterior median simulation and the ribbon represents the 2.5%-97.5% posterior quantiles in the median. In both plots, the modelled % infected midguts values were determined by calculating the probability the titer exceeds the detection threshold, specified by Eq (8).
(TIFF)

**S4 Fig. Model sensitivities: Parameters affecting midgut invasion.** Each column shows the impact of varying a model parameter on the resultant dynamics. The top / bottom rows show the viral dynamics in the midgut / legs. The parameters are varied by multiplying the estimated posterior mean value by a multiplier, shown by the colouring. The black dashed lines shows the model simulations at the posterior mean values.
(TIFF)

**S5 Fig. The order of infectious blood meals affects the rate of dissemination.** The line colouring shows results for three feed types: mosquitoes fed only at time 0 ("single"); mosquitoes fed an infected blood meal at time 0 followed by an uninfected blood meal 3 DPI ("double: infectious first"); and mosquitoes fed an uninfected blood meal at time 0 followed by an infected blood meal 3 DPI ("double: uninfectious first"). Panel A / B shows the modelled proportion of the midguts / legs positive to infection over time across the three feeds. The dose of initial infection corresponds to a concentration of 0.5 (the same as in Fig 6). In both plots, the modelled % infected values were determined by calculating the probability the titer exceeds the detection threshold.
(TIFF)

**S6 Fig. Model sensitivities: Single versus double feeding.** Each row corresponds to a parameter; each column to a multiple of the posterior mean of that parameter. The lines show the model-predicted infection profiles for mosquitoes that were either singly fed or those fed an infectious blood meal at 0DPI, followed by a second noninfectious blood meal at 3DPI. Each

panel corresponds to model predictions for a single parameter being varied in isolation. (TIFF)

**S7 Fig. Time-varying Sobol' indices.** Each column shows the modelled output considered; each row shows results corresponding to the labelled parameter. The coloured lines correspond to the first order ('Si') and total order ('Ti') Sobol' indices, and the ribbons show the 95% confidence intervals in these quantities as calculated via the *sensobol* R package [76]. Sensitivities were estimated using $2^{10}$ model simulations sampled across a subset of parameter space using Quasi-Random numbers (the default package option). The subset of parameter space considered was the hypercube bounded below at 2/3 times the posterior median estimates and above at 3/2 times these estimates. Note, the level of virus in the lumen decays towards zero in the first three days following a blood feed (here mosquitoes were singly fed), so we remove the subsequent sensitivities from the plot to avoid error due to numerical underflow.
(TIFF)

**S8 Fig. Prior sensitivities.** The vertical axis displays the parameter whose prior was perturbed by multiplying the scale parameter by a factor of 5: for example, $a \sim$ normal(0, 10) becomes $a \sim$ normal(0, 50). Within a given row, the colouring shows the effect of these changes to the prior on the posterior distributions, as quantified by (A) the Wasserstein distance and (B) the ratio of the posterior mean estimates between the posterior distribution estimated using the priors shown in Table S1 Table and the perturbed ones. The perturbed posteriors were estimated using Pareto-smoothed importance sampling via the *adjustr* package [77]. In (B), the patterning indicates those cases where the posterior mean from the perturbed priors was within 10% of the original estimate. The rows have been ordered according to the Pareto-k value (larger values at the top). Note that the first three rows ($k_{mh}$: $\gamma$) have Pareto-k values above 0.7, indicating unreliable estimates [78].
(TIFF)

## Author Contributions

**Conceptualization:** Rebecca M. Johnson, Helen M. Byrne, Philip M. Armstrong, Douglas E. Brackney, Ben Lambert.

**Data curation:** Rebecca M. Johnson, Philip M. Armstrong, Douglas E. Brackney, Ben Lambert.

**Formal analysis:** Rebecca M. Johnson, Isaac J. Stopard, Helen M. Byrne, Philip M. Armstrong, Douglas E. Brackney, Ben Lambert.

**Funding acquisition:** Philip M. Armstrong, Douglas E. Brackney.

**Investigation:** Rebecca M. Johnson, Isaac J. Stopard, Helen M. Byrne, Philip M. Armstrong, Douglas E. Brackney, Ben Lambert.

**Methodology:** Rebecca M. Johnson, Isaac J. Stopard, Helen M. Byrne, Philip M. Armstrong, Douglas E. Brackney, Ben Lambert.

**Project administration:** Philip M. Armstrong, Douglas E. Brackney, Ben Lambert.

**Resources:** Rebecca M. Johnson, Philip M. Armstrong, Douglas E. Brackney, Ben Lambert.

**Software:** Isaac J. Stopard, Philip M. Armstrong, Ben Lambert.

**Supervision:** Helen M. Byrne, Philip M. Armstrong, Douglas E. Brackney, Ben Lambert.

**Validation:** Isaac J. Stopard, Helen M. Byrne, Philip M. Armstrong, Douglas E. Brackney, Ben Lambert.

**Visualization:** Rebecca M. Johnson, Isaac J. Stopard, Helen M. Byrne, Philip M. Armstrong, Douglas E. Brackney, Ben Lambert.

**Writing – original draft:** Rebecca M. Johnson, Isaac J. Stopard, Helen M. Byrne, Philip M. Armstrong, Douglas E. Brackney, Ben Lambert.

**Writing – review & editing:** Rebecca M. Johnson, Isaac J. Stopard, Helen M. Byrne, Philip M. Armstrong, Douglas E. Brackney, Ben Lambert.

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
