## [Decision Letter · Decision Letter 0]

3 Nov 2023

Dear Dr Lambert,

Thank you very much for submitting your manuscript "Investigating the dose-dependency of the midgut escape barrier using a mechanistic model of within-mosquito dengue virus population dynamics" for consideration at PLOS Pathogens. As with all papers reviewed by the journal, your manuscript was reviewed by members of the editorial board and by several independent reviewers. In light of the reviews (below this email), we would like to invite the resubmission of a significantly-revised version that takes into account the reviewers' comments.

We cannot make any decision about publication until we have seen the revised manuscript and your response to the reviewers' comments. Your revised manuscript is also likely to be sent to reviewers for further evaluation.

Sincerely,

Louis Lambrechts

Guest Editor

PLOS Pathogens

Alexander Gorbalenya

Section Editor

PLOS Pathogens

Kasturi Haldar

Editor-in-Chief

PLOS Pathogens

orcid.org/0000-0001-5065-158X

Michael Malim

Editor-in-Chief

PLOS Pathogens

orcid.org/0000-0002-7699-2064

Reviewer's Responses to Questions

**Part I - Summary**

Reviewer #1: This paper looks at dose dependence of within-mosquito DENV kinetics through a combination of experimental and mathematical work. The manuscript addresses a gap in knowledge regarding this topic, and both the experimental and computational experiments support their overall conclusions. However, there are some minor concerns regarding the organization of the output.

Reviewer #2: Please see attached word document

Reviewer #3: The paper presents a novel mechanistic within-vector model of dengue infection in Aedes mosquitoes. The model is fitted to data on dengue infection, including single and double blood feeds. The topic is very relevant and not extensively examined for viral infections. Overall, the paper is well written, with clear justification of modeling choices, and acknowledgement of limitations. While the model is simplistic, it is an excellent first step with a nice road map of ways to improve the model (e.g. stochasticity). In particular, the careful attention to detail on the model formulation, providing all the necessary components for replication, is rare and is much appreciated.

**Part II – Major Issues: Key Experiments Required for Acceptance**

Reviewer #1: None

Reviewer #2: Please see attached word document

Reviewer #3: No major issues.

**Part III – Minor Issues: Editorial and Data Presentation Modifications**

Reviewer #1: Introduction/Discussion

See Christofferson et al. “Characterizing the likelihood of dengue emergence and detection in naïve populations” for an example of DENV dose-dependence and transmission.

See Veronesi et al. “Estimating the Impact of Consecutive Blood Meals on Vector Competence of Aedes albopictus for Chikungunya Virus” for additional observations of a second blood meal not affecting CHIKV.

M&M:

“… this provides semi-continuous data…” Three data points are longitudinal, but was a curve fit to these data to achieve semi-continuous?

What was the titer of the initial bloodmeal so that the dilutions have a titer associated with them? This is not clear. If this does address this point, consider moving this above.

Page 7: Mayton et al. “A method for repeated, longitudinal…” might also be relevant.

Results:

“At 3 DPI, mosquitoes feeding on the 1:1 viral dilution show higher midgut DENV titers but this early lead is

transient, and the viral titer in the midgut layer is largely independent of the initial blood meal dose by 8 DPI.” – provide a statistical test for this statement.

In text, the reference to dashed lines for Figure 5 is a bit confusing, consider saying “within the vertical dashed lines”

Line 435-437 – this paper shows a single timepoint (not the latest) at which dissemination was higher in Ae. albopictus, so “substantially” may not be warranted

Lines 430-431 – this section makes a lot of supposition about mutations. Other external forces affect EIP and vector competence, including temperature which might affect mosquito metabolism rather than viral kinetics directly… Here you are modeling a phenotype, there is no evidence to suggest the A226V mutation affects any of the parameters being toggled here. That’s not to say that this computational experiment is not interesting or informative, but I worry putting it into an assumed mechanism from a mutation will be overinterpreted and mis-cited in future. Consider softening the association with mutations and just put it in the context of alternative phenotypes.

Reviewer #2: Please see attached word document

Reviewer #3: - It is unusual to call the model “time-dependent” as done in the methods and discussion. The equations are autonomous, with no explicit dependence on time. It would be more accurate to call the model a system of autonomous ODEs and remove “time-dependent”

- Why is a Hill coefficient of 2 chosen for the rate that the virus invades the midgut? This typically indicates cooperation. This is one of the few modeling choices not discussed.

- In general, all figures could do with larger font.

- In Figures 2 and 3, the colors can be hard to distinguish (especially for the points). Consider using different marker symbols.

- In Figure 5 B and C, the font is small to the point of being difficult to read.

- In Figures 6, S1, S4, S5, S6, S7, write out what DPI means.

- In Figure 7, label the axis with words in addition to symbols. The symbols are not mentioned in the caption either.

- In Figure S2, what does *** indicate?

- The final paragraph seems out of place. While this reviewer agrees with the sentiment, it’s not clear it’s appropriate at this point in the manuscript.

PLOS authors have the option to publish the peer review history of their article (what does this mean?). If published, this will include your full peer review and any attached files.

Reviewer #1: No

Reviewer #2: No

Reviewer #3: No
---

## [Decision Letter · Decision Letter 1]

9 Jan 2024

Dear Dr Lambert,

Thank you very much for submitting your manuscript "Investigating the dose-dependency of the midgut escape barrier using a mechanistic model of within-mosquito dengue virus population dynamics" for consideration at PLOS Pathogens. As with all papers reviewed by the journal, your manuscript was reviewed by members of the editorial board and by several independent reviewers. The reviewers appreciated the attention to an important topic. Based on the reviews, we are likely to accept this manuscript for publication, providing that you modify the manuscript according to the review recommendations.

Most of the previous comments have been satisfactorily addressed. The stylistic choice to keep the manuscript as is, rather than change to a purely results section without interpretation, is fine. However, Reviewer #2 has pointed out a couple of minor corrections that remain to be made.

Sincerely,

Louis Lambrechts

Guest Editor

PLOS Pathogens

Alexander Gorbalenya

Section Editor

PLOS Pathogens

Kasturi Haldar

Editor-in-Chief

PLOS Pathogens

orcid.org/0000-0001-5065-158X

Michael Malim

Editor-in-Chief

PLOS Pathogens

orcid.org/0000-0002-7699-2064

Most of the previous comments have been satisfactorily addressed. The stylistic choice to keep the manuscript as is, rather than change to a purely results section without interpretation, is fine. However, Reviewer #2 has pointed out a couple of minor corrections that remain to be made.

Reviewer Comments (if any, and for reference):

Reviewer's Responses to Questions

**Part I - Summary**

Reviewer #1: The revision clarified some of ambiguity in parts of the manuscript.

Reviewer #2: No further comments

Reviewer #3: The paper presents a novel mechanistic within-vector model of dengue infection in Aedes mosquitoes. The model is fitted to data on dengue infection, including single and double blood feeds. The topic is very relevant and not extensively examined for viral infections, as evidenced by another recently published paper addressing a similar topic. Overall, the paper is well written, with clear justification of modeling choices, and acknowledgement of limitations.

**Part II – Major Issues: Key Experiments Required for Acceptance**

Reviewer #1: None

Reviewer #2: There are no major issues

Reviewer #3: None.

**Part III – Minor Issues: Editorial and Data Presentation Modifications**

Reviewer #1: All comments were addressed.

Reviewer #2: I still think the manuscript could be made somewhat more concise by restricting the results section just to the results and therefore easier to digest, but accept that this is just a difference in opinion concerning style.

From line 48 statement is still not correct:

While some studies measure dissemination as the proportion of all blood-fed mosquitoes with disseminated infection [21, 22], this makes it impossible to separate out the impact of changes to the bloodmeal titer on infection and subsequent dissemination. Thus we measured dissemination as the proportion of mosquitoes with infected midguts which have disseminated infection.

Both studies measure and present results on the proportion of infected mosquitoes that have a disseminated infection, which makes it possible to separate out the impact of changes.

Fig 1. D. ‘other tissues’ should be h(t) not m(t)

Reviewer #3: The authors have addressed all concerns of this reviewer during revision.

PLOS authors have the option to publish the peer review history of their article (what does this mean?). If published, this will include your full peer review and any attached files.

Reviewer #1: No

Reviewer #2: No

Reviewer #3: No

Figure Files:

Data Requirements:

Reproducibility:

References:

---

## [Editor Report · Decision Letter 2]

16 Jan 2024

Dear Dr Lambert,

We are pleased to inform you that your manuscript 'Investigating the dose-dependency of the midgut escape barrier using a mechanistic model of within-mosquito dengue virus population dynamics' has been provisionally accepted for publication in PLOS Pathogens.

Best regards,

Louis Lambrechts

Guest Editor

PLOS Pathogens

Alexander Gorbalenya

Section Editor

PLOS Pathogens

Michael Malim

Editor-in-Chief

PLOS Pathogens

orcid.org/0000-0002-7699-2064

Editor Comments:

All final comments have been addressed, congratulations to the authors on a nice study !

---

## [Editor Report · Acceptance letter]

25 Mar 2024

Dear Dr Lambert,

We are delighted to inform you that your manuscript, "Investigating the dose-dependency of the midgut escape barrier using a mechanistic model of within-mosquito dengue virus population dynamics," has been formally accepted for publication in PLOS Pathogens.

Best regards,

Michael Malim

Editor-in-Chief

PLOS Pathogens

orcid.org/0000-0002-7699-2064